# Solidification Materials and Technology for Solid Self-Emulsifying Drug Delivery Systems

**DOI:** 10.3390/ph18101550

**Published:** 2025-10-15

**Authors:** Kyungho Baek, Sung Giu Jin

**Affiliations:** 1College of Pharmacy, Hanyang University, Ansan 15588, Republic of Korea; 2College of Pharmacy, Dongguk University, Goyang 10326, Republic of Korea

**Keywords:** solid self-emulsifying drug delivery system, solid carrier, solidification method

## Abstract

The low aqueous solubility of many new drug candidates, a key challenge in oral drug development, has been effectively addressed by liquid self-emulsifying drug delivery systems (SEDDS). However, the inherent instability and manufacturing limitations of liquid formulations have prompted significant research into solid SEDDS. This review provides a comprehensive analysis of the recent advancements in solid SEDDS, focusing on the pivotal roles of solid carriers and solidification techniques. We examine a wide range of carrier materials, including mesoporous silica, polymers, mesoporous carbon, porous carbonate salts, and clay-based materials, highlighting how their physicochemical properties can be leveraged to control drug loading, release kinetics, and in vivo performance. We also detail the various solidification methods, such as spray drying, hot melt extrusion, adsorption, and 3D printing, and their impact on the final product’s quality and scalability. Furthermore, this review explores applications of solid SEDDS, including controlled release, mucoadhesive technology, and targeted drug delivery, as well as the key commercial challenges and future perspectives. By synthesizing these diverse aspects, this paper serves as a valuable resource for designing high-performance solid SEDDS with enhanced stability, bioavailability, and functional versatility.

## 1. Introduction

One of the significant challenges in the development of orally administered drugs is the low aqueous solubility of drug candidates [1]. A considerable number of new chemical entities (40–70%) in the drug development pipeline possess low solubility, which often leads to poor gastrointestinal absorption and, ultimately, limited therapeutic efficacy [2,3]. These poorly water-soluble drugs fall into Class II (low solubility, high permeability) and Class IV (low solubility, low permeability) of the Biopharmaceutics Classification System (BCS) [4,5]. Improving the bioavailability of such drugs has been a persistent challenge for successful drug products. To overcome this issue, lipid-based drug delivery systems (LDDS) have been extensively studied as an effective strategy. LDDS have the potential to enhance the oral bioavailability of drugs by increasing drug solubility, improving intestinal permeability, and bypassing first-pass metabolism via lymphatic transport [6,7,8,9].

Among the various LDDS, self-emulsifying drug delivery systems (SEDDS) have garnered significant attention and are a particularly promising system. SEDDS are isotropic liquid mixtures composed of oil, surfactant, and co-surfactant. Upon oral administration, when they come into contact with the aqueous environment of the gastrointestinal tract, they spontaneously form a fine oil-in-water emulsion [10,11]. This process occurs with only the mild agitation provided by gastrointestinal motility. The resulting fine emulsion droplets increase the surface area of the drug, accelerate drug dissolution, and facilitate the rapid movement of the solubilized drug across the gastrointestinal mucosa. Consequently, SEDDS have been shown to significantly improve drug solubilization, enhance absorption, and thereby increase bioavailability [12].

Despite their advantages, the liquid formulation of SEDDS presents several limitations. The majority of commercially available SEDDS products (e.g., Sandimmune^®^, Norvir^®^) are in liquid form, which can lead to the following problems during manufacturing and storage: (1) Low stability—Liquid formulations are prone to physical and chemical instability over time, such as precipitation, phase separation, or capsule leakage [13,14]. (2) Manufacturing and handling difficulties—The capsule-filling process for liquid formulations is relatively complex and may limit large-scale production [15]. (3) Limited formulation versatility—Typically confined to soft or hard gelatin capsules, it is difficult to adjust the formulation for diverse therapeutic purposes, which can lead to poor patient compliance [16].

To overcome the drawbacks of liquid SEDDS while maintaining their drug delivery efficiency, there has been a surge of research into solid SEDDS. Solid SEDDS are systems in which the core components of conventional liquid SEDDS are solidified using various solidification techniques, such as spray drying, melt extrusion, or freeze-drying, with the help of solid carriers [17,18,19].

Solid SEDDS combine the advantages of liquid SEDDS with the benefits of solid dosage forms: (1) Improved physical and chemical stability—They resolve the instability issues of liquid formulations, maintaining stability without drug precipitation or degradation during long-term storage [14]. (2) Versatile formulation possibilities—They can be manufactured into various forms, such as tablets, pellets, or powders, which enhances patient convenience and opens up possibilities for developing functional dosage forms like sustained-release formulations [20]. (3) Efficient manufacturing process—Solid SEDDS can leverage existing solid-dosage manufacturing lines, making the process highly efficient and more suitable for large-scale commercial production [21].

This review aims to provide a comprehensive analysis of the advancements in solid self-emulsifying drug delivery systems. We will focus on the solidification materials and techniques central to solid SEDDS formulations, highlighting various strategies to overcome the limitations of liquid SEDDS and optimize drug performance.

## 2. Liquid SEDDS

To successfully formulate as SEDDS, it is essential to select appropriate excipients based on the physicochemical properties of the drug and to optimize their proportions. SEDDS are primarily composed of three core components: oil, surfactant, and co-surfactant. These components do more than simply mix; they play a complex, synergistic role in maximizing drug solubilization and absorption efficiency [22].

### 2.1. Lipids/Oils

Oils serve as the primary solvent for the drug in SEDDS formulations and form the core of the emulsion’s oil phase after oral administration and subsequent self-emulsification. The most critical criterion for oil selection is the drug’s solubility. The drug must be sufficiently dissolved in the oil phase to remain stable and avoid precipitation, even after dilution in the gastrointestinal tract [23].

Oils are broadly classified into medium-chain triglycerides (MCTs) (C6–C12) and long-chain triglycerides (LCTs) (over C12), based on the fatty acid chain length. MCTs: MCTs are characterized by low viscosity and high solvent capacity. They possess superior oxidative stability compared to LCTs and exhibit excellent self-emulsifying and solubilizing abilities [24]. They also tend to be rapidly hydrolyzed in the gastrointestinal tract, which can accelerate drug absorption. However, they are primarily absorbed via portal circulation, which may limit their ability to bypass first-pass metabolism through lymphatic transport. Representative MCT oils include Capryol^®^ 90, Captex^®^ 300, and Labrafac^®^ CC [25]. LCTs: LCTs are advantageous for promoting drug transport via the lymphatic system, which allows them to bypass hepatic first-pass metabolism. This is a crucial mechanism for maximizing drug oral bioavailability [26]. However, LCTs have the disadvantage of higher viscosity and are relatively more difficult to emulsify. Examples of LCTs include Maisine^®^-35, Lauroglycol^®^ 90, and Peceol^®^ [27].

Natural and Modified Oils: While natural oils like olive oil and corn oil are safe, their low solubility for lipophilic drugs can be a limiting factor. Recent research has seen the widespread use of modified vegetable oils, particularly semi-synthetic lipids containing various mono- and diglycerides, which have improved emulsifying properties and drug solubility. These also provide amphiphilic characteristics, allowing them to act similarly to surfactants [22].

### 2.2. Surfactants

Surfactants play a decisive role in the self-emulsifying ability and stability of SEDDS formulations. They lower the interfacial tension between the oil and water phases, minimizing the free energy required for emulsion formation, and stabilize the resulting nano-emulsion droplets to prevent their aggregation [28]. Due to their low toxicity and stability over a wide pH range, non-ionic surfactants are predominantly used in SEDDS [29]. Furthermore, surfactants do more than act as emulsifiers; they also have various mechanisms to enhance drug absorption. They can promote passive transport by increasing the fluidity of the intestinal epithelial cell membrane or enhance intestinal absorption by inhibiting drug efflux transporters, such as P-glycoprotein [30].

The selection of a surfactant must satisfy both drug solubility and oil emulsification efficiency. The effective concentration of surfactants in SEDDS typically ranges from 30–60%, but high concentrations can cause gastrointestinal irritation and cytotoxicity. Therefore, it is advisable to use the lowest possible concentration while maintaining the desired performance [21]. Representative non-ionic surfactants used in SEDDS include polyoxyl castor oils (Kolliphor^®^ EL, RH40), polysorbates (Tween^®^ 20, 80), macrogolglycerides (Labrasol^®^), polyoxyl 15 hydroxystearate (Kolliphor^®^ HS 15), and vitamin E TPGS; their relatively high HLB values (typically > 12) promote rapid o/w self-emulsification and stable nanoemulsions.

### 2.3. Co-Surfactants/Co-Solvents

Co-surfactants are auxiliary components that work synergistically with surfactants to improve emulsification efficiency and enhance the thermodynamic stability of the formulation [31]. Co-surfactants increase the fluidity of the oil–water interface, thereby more effectively reducing interfacial tension. This enables the formation of smaller, more uniform nano-emulsion droplets and improves the stability of the formulation by penetrating between surfactant molecules to prevent liquid crystal formation [32]. Hydrophilic substances such as short-chain alcohols (e.g., ethanol, isopropyl alcohol) or glycols (e.g., propylene glycol) are commonly used. These are suitable for oral administration and enhance the miscibility between the oil and surfactant within the formulation. Although co-surfactants can contribute to enhanced drug solubility, they can also easily migrate to the aqueous medium, potentially causing drug precipitation [33]. Furthermore, volatile co-solvents can lead to capsule leakage, so their amount should be minimized to ensure the formulation’s stability. Representative examples include Transcutol^®^ HP and Glycofurol^®^ [34].

### 2.4. Characteristics of Suitable Drugs for SEDDS

SEDDS are not a universal solution suitable for all drugs. This formulation is primarily effective for poorly water-soluble drugs corresponding to BCS Class II and Class IV [25]. The optimal performance of a drug in an SEDDS formulation is observed when it possesses the following characteristics: (1) Lipophilicity: Drugs with a logP value greater than 5 are known to be more suitable for SEDDS formulations due to their higher solubility in lipid-based excipients [35]. (2) Low Dose: The lower the drug dose, the easier it is to effectively dissolve it in the mixture of oil, surfactant, and co-surfactant [36].

## 3. Solid SEDDS

Conventional liquid SEDDS face several challenges, including the high cost associated with soft gelatin capsules and stability issues stemming from capsule leakage of oil components and their interactions with the capsule shell [37]. Furthermore, drug precipitation, poor portability, and a limited range of formulation options have constrained patient compliance and commercial large-scale production [16]. To overcome these drawbacks while retaining the advantages of liquid SEDDS, solid SEDDS have emerged as a promising alternative [38].

Solid SEDDS are created by incorporating liquid SEDDS into a solid carrier or by solidifying them into solid dosage forms like powders, pellets, capsules, or tablets using various solidification techniques [39,40]. Solid SEDDS preserve the excellent drug solubility and bioavailability-enhancing effects of liquid SEDDS while offering several significant advantages: (1) Improved Stability: They exhibit superior physical and chemical stability, effectively preventing drug precipitation or oxidative degradation of lipid components during long-term storage [41,42]. (2) Ease of Manufacturing and Cost-Effectiveness: Solid SEDDS can be manufactured using standard solid dosage form production equipment, which facilitates large-scale production (scale-up) and reduces manufacturing costs [43]. (3) Formulation Versatility: They can be prepared in various forms, such as tablets and pellets, in addition to capsules. This not only enhances patient compliance but also allows for the incorporation of controlled drug-release functionalities [20]. (4) Enhanced Safety: The solidification process can help reduce the amount of high-concentration surfactants, thereby lowering the potential risk of toxicity, such as gastrointestinal irritation [20].

Recent research trends in solid SEDDS have moved beyond simply solidifying the system to a deeper analysis of how the choice of solid carrier material impacts the performance of the formulation. While earlier work emphasized manufacturability aspects such as re-dispersion efficiency, current approaches are now leveraging the physicochemical properties of solid carriers to further enhance SEDDS performance [16,44]. For example, specific solid carriers can offer additional benefits, such as controlling the drug release rate and modulating bioavailability [25,45]. A thorough understanding of the diverse solid carrier materials for solidification and their functional roles is essential for designing high-performance solid SEDDS.

## 4. Role and Types of Solid Carriers for Solid SEDDS

The selection of a solid carrier is crucial for the successful development of solid SEDDS, as it effectively holds the liquid components and determines the physical properties of the final formulation (Figure 1). An ideal solid carrier should possess the following characteristics: (1) High liquid-loading capacity: It must have high porosity and surface area to sufficiently absorb and stably hold the liquid SEDDS [46]. (2) Excellent physical properties: In powder form, it should have good flowability, and when tableted, it must possess sufficient mechanical strength [47]. (3) Controlled redispersion properties: The hydrophilicity and porosity of the carrier play a vital role in controlling the drug release rate and biopharmaceutical performance in vivo [48].

Solid carriers used in solid SEDDS can be broadly classified into water-soluble and water-insoluble carriers. Water-soluble carriers, such as polymers, polysaccharides, and protein-based materials (e.g., dextran, cellulose derivatives), dissolve easily in water, promoting the rapid release of the incorporated drug [49,50]. Water-insoluble carriers, such as porous silica (e.g., fumed silica) and silicate-based carriers, do not dissolve in water but can absorb large amounts of liquid SEDDS due to their high porosity and large surface area. These are particularly useful for controlling the drug release profile of the formulation [51].

Recent research has moved beyond simply finding suitable carriers for solidification to focusing on the intelligent selection of solid carriers to control drug release and bioavailability by leveraging their physicochemical properties. As will be explored in the subsequent section on solidification carriers, the choice of a solid carrier for solid SEDDS performs more than just a solidification function. The physical structure, surface chemistry, and hydrophilicity–hydrophobicity characteristics of the carrier critically influence the drug release rate, redispersion efficiency, and, ultimately, the in vivo drug absorption rate [52]. Therefore, to design an optimal solid SEDDS, the type and characteristics of the solid carrier must be carefully considered based on the drug’s properties and the desired therapeutic effect. In particular, silica and silicate-based carriers, as well as inorganic mesoporous materials, can enhance the solidification of solid SEDDS due to their hydrophobic properties, allowing for controlled drug release. Conversely, polymer-based and carbohydrate-based carriers, while exhibiting lower encapsulation or immobilization efficiency due to their hydrophilic properties, offer the distinct advantage of enhancing drug absorption rates. Therefore, when designing solid SEDDS, the necessary solid carrier must be selected based on the characteristics of the target product.

### 4.1. Silica and Silicate-Based Carriers

Silica and silicate-based materials have a long history of use in pharmaceutical formulations due to their large surface area and high adsorption capacity [53,54]. They enhance oral absorption by stabilizing drug molecules in an amorphous state, increasing drug wettability via their hydrophilic surfaces, and promoting supersaturated drug dissolution by acting as precipitation inhibitors [55]. Representative silica-based carriers include fumed silica nanoparticles (Aerosil^®^) and magnesium aluminum silicate (Neusilin^®^) [56].

Aerosil^®^ and Neusilin^®^, due to their large surface area, can load a significant amount of lipid, creating free-flowing powders and demonstrating high drug bioavailability upon redispersion. In particular, the surface chemistry of the carrier (e.g., ionization based on pH) can influence the drug release rate through electrostatic interactions with charged drug molecules [57]. Mesoporous silicates such as Aerosil^®^ and Neusilin^®^ have shown notable potential as solid carriers for SEDDS. For instance, SEDDS adsorbed onto Neusilin^®^ US2 at a 1:1 *w*/*w* ratio produced tablets with high mechanical strength and immediate drug release, provided that gelation did not occur upon hydration [58,59]. Other reports have noted delayed lipid dispersion and reduced absorption due to strong surfactant–carrier interactions, particularly with Neusilin^®^ [60]. To overcome these issues, precoating Neusilin^®^ US2 with hydrophilic polymers such as polyvinylpyrrolidone has been proposed to minimize surfactant binding and improve release performance [61]. Additionally, silica–lipid hybrid systems using Aerosil^®^ 300 and Syloid^®^ 244 have demonstrated that tailoring silica geometry and pore structure can enhance supersaturation stability and in vivo bioavailability, achieving greater oral absorption than conventional formulations [62]. Silica-based carriers can also influence gastrointestinal lipid digestion by modulating lipase access to encapsulated lipids. Depending on surface hydrophilicity and pore accessibility, formulations may accelerate or delay lipid digestion, providing control over absorption kinetics and, in some cases, enhancing lymphatic transport [63].

Silica and silicate-based materials are the most frequently used; however, their hydrophobic nature presents the risk of reduced initial dissolution and drug release when they are incorporated in large quantities due to potential drug binding. Overall, optimal design and selection of solid carriers by considering surface properties, porosity, and compatibility with lipid-based excipients are crucial for maximizing the pharmaceutical performance of solid SEDDS.

### 4.2. Polymer-Based Carriers

Polymer-based materials such as hydroxypropyl methylcellulose (HPMC), sodium carboxymethylcellulose (Na-CMC), poloxamers, and cyclodextrins are used as solid SEDDS carriers due to their ability to adsorb/immobilize the liquid SEDDS and to control drug release [64]. These polymer carriers can impart controlled-release characteristics to solid SEDDS, which is beneficial for preventing drug precipitation and extending the duration of action for drugs with short half-lives [65,66].

Various polymeric carriers have been shown to enhance the performance of solid SEDDS. For example, HPMC was used in a spray-dried solid SEDDS containing the oily drug and calcium silicate, producing nano-sized droplets (<300 nm) with improved thermal stability and a threefold increase in oral bioavailability compared to conventional soft capsules [67]. Na-CMC was identified as an optimal hydrophilic carrier in a nanoparticle screening system for aceclofenac, where solid SEDDS and self-nanoemulsifying granule systems achieved approximately 1500-fold higher aqueous solubility and significantly improved bioavailability [68]. Poloxamer^®^ 188 has been reported to function as both a solidifying and emulsifying agent in SEDDS formulations. Its amphiphilic and thermoresponsive properties enabled stable lipid incorporation, spontaneous emulsification, and rapid melting at body temperature, which facilitated improved dissolution and drug release [69,70]. Hydroxypropyl-β-cyclodextrin acted as both a hydrophilic carrier and solubilizer, forming spherical solid SEDDS particles (~120 nm) with amorphous drug dispersion and markedly higher solubility than silica-based systems [71]. Chitosan–EDTA microparticles, prepared by solvent evaporation or spray drying, showed excellent oil adsorption capacity and structural stability for solidifying liquid SEDDS. Optimized compositions demonstrated enhanced zeta potential, amorphous structure, and significantly improved drug loading and release, particularly for thermolabile drugs such as amphotericin B [72,73]. Similarly, Soluplus^®^ and Kollicoat^®^ IR, two grafted copolymers, were evaluated as carriers for solid dispersions of arteether. Soluplus^®^ showed superior saturation solubility, powder flow, and in vitro dissolution (~89%) when processed by spray drying and encapsulated in enteric-coated capsules [74]. In another approach, hydrophobic ionic complexes formed between anionic drugs and cationic Eudragit^®^ polymers (RS, RL, E) were incorporated into SEDDS to achieve sustained release. The release kinetics were influenced by the polymer-to-drug charge ratio and the type of Eudragit^®^ RL, producing the most prolonged release profile [75].

Polymer-based materials, as mentioned above, possess many advantages. However, they also have the disadvantage of requiring large quantities to adsorb and solidify liquid SEDDS. Therefore, the large amount of polymer required for solidification poses the risk of delayed disintegration and increased viscosity, which delays re-emulsification and leads to system instability. Overall, polymer-based carriers offer broad functional versatility in solid SEDDS design. By improving solubility, drug loading, emulsification, and controlled release, they provide an effective means of tailoring formulation performance according to drug properties and therapeutic goals

### 4.3. Inorganic Mesoporous Materials

#### 4.3.1. Mesoporous Carbon

Mesoporous carbon boasts a remarkably high specific surface area, and a large pore volume, surpassing that of mesoporous silica. This allows for a significant increase in the drug loading capacity of solid SEDDS when mesoporous carbon is used as a solid carrier [76].

Furthermore, its hydrophobic surface properties enhance its affinity for lipid excipients, contributing to the formation of a stable solid formulation. Mesoporous carbon can stabilize drug molecules through weak physical adsorption interactions, which in turn helps to suppress burst release and induce a sustained release profile. This feature allows for fine-tuning of the drug dissolution profile, preventing drug precipitation and helping to optimize the absorption window. Recent studies have highlighted the potential of mesoporous carbon as a solid carrier in lipid-based self-emulsifying systems. For example, nimodipine-loaded mesoporous carbon nanoparticles coated with lipid bilayer shells achieved high drug loading (~27%) and sustained release for up to 18 h, resulting in a 2.14-fold increase in oral bioavailability compared with conventional immediate-release tablets [77]. Similarly, ordered mesoporous carbon with a surface area of 1174 m^2^/g enabled incorporation of fenofibrate in an amorphous, highly dispersed state, leading to over 80% drug release within 1 h and approximately 1.5-fold higher bioavailability than crystalline formulations [78]. Comparative studies of hollow and solid mesoporous carbon nanoparticles further showed that both structures achieved high drug-loading efficiency, significantly improved dissolution, and maintained long-term physical stability of poorly soluble drugs under accelerated storage conditions [79].

Overall, mesoporous carbon carriers can preserve the self-emulsifying performance of solidified formulations while improving drug loading, enabling controlled release, and enhancing the oral bioavailability of poorly water-soluble compounds. These attributes make them promising candidates for advanced solid SEDDS development.

#### 4.3.2. Porous Carbonate Salts

Due to their pH-sensitive nature, porous carbonate salts are promising materials for targeted drug delivery, in addition to being used as solidification carriers for solid SEDDS [63]. Porous calcium carbonate possesses a high specific surface area and excellent biocompatibility. However, its most significant limitation is a phase transition to non-porous calcite in the presence of water. During this process, the loaded drug can recrystallize, leading to a decrease in its dissolution rate. There is also a risk of immediate drug release (dose dumping) and precipitation in the aqueous environment of the gastrointestinal tract [80]. Mesoporous magnesium carbonate is a new mesoporous material that overcomes the instability issues of porous calcium carbonate, demonstrating high stability in an aqueous environment. It has a specific surface area of up to 800 m^2^g^−1^, and it can improve the dissolution rate by stabilizing the drug in a molecularly dispersed amorphous state. Furthermore, its drug release profile can be finely controlled by adjusting the particle size [81].

Recent studies have applied porous carbonate salts as solid carriers for self-nanoemulsifying formulations, leveraging their high surface area and favorable drug–carrier interactions. Functionalized calcium carbonate (FCC) effectively adsorbed supersaturated SEDDS containing carvedilol, maintaining nanodroplet size and polydispersity comparable to the liquid formulation. The resulting solid SEDDS achieved rapid drug release (>80% within 5 min) and retained the physical stability of the supersaturated drug load for up to 10 weeks under ambient storage, demonstrating FCC’s potential as a robust and biocompatible carrier [80]. Similarly, mesoporous magnesium carbonate has been reported to improve the oral bioavailability of poorly water-soluble drugs such as celecoxib by maintaining them in a non-crystalline state and facilitating rapid release upon administration. Owing to its high surface area and tunable porous structure, mesoporous magnesium carbonate enables efficient drug loading and stabilization, making it a promising carrier for solid SEDDS to prevent recrystallization and enhance dissolution performance [82].

#### 4.3.3. Clay-Based Materials

Clay-based materials have a long history of safe use as pharmaceutical excipients and can control drug release through their unique layered structure and ion-exchange capabilities [83].

Montmorillonite: This clay material has negatively charged layers that can adsorb cationic drugs, thereby controlling their release. However, its limitation is that the ion exchange is an equilibrium reaction, which can lead to incomplete drug release [56].

Layered Double Hydroxides (LDHs): In contrast to montmorillonite, LDHs have a positive charge. They control drug release by intercalating anionic drugs between their layers. A key characteristic is their rapid dissolution in acidic environments, such as gastric acid. While this can be a drawback for oral drug delivery, it can also be leveraged to design targeted systems for rapid drug release at specific sites [84].

Recent studies have demonstrated the potential of clay-based carriers for solid SEDDS. Montmorillonite and laponite clays have been applied in a dual-loading approach, combining drug intercalation within clay layers and encapsulation in a lipid phase. This strategy increased drug loading to as much as 20.3% while maintaining favorable dissolution profiles. Dual-loaded montmorillonite achieved oral bioavailability comparable to its liquid counterpart, confirming the utility of layered silicate clays as effective carriers [85]. LDHs have also been developed as intercalation systems for hydrophobic molecules, with sodium dodecyl sulfate modification enabling efficient encapsulation and stabilization of poorly soluble drugs. By offering tunable interlayer chemistry and pH-responsive release, LDH-based carriers provide a promising platform for solid SEDDS to enhance drug loading and achieve controlled release of lipophilic compounds [86,87].

In general, inorganic mesoporous-based materials have hydrophobic characteristics, which poses a risk of reduced initial dissolution and drug release due to potential drug binding when mixed in large quantities. Designing an optimal ratio that considers the characteristics of each inorganic mesoporous-based material is crucial for maximizing the performance of solid SEDDS.

### 4.4. Carbohydrate-Based Materials

Carbohydrate-based materials are highly biocompatible and biodegradable, and they can provide solid SEDDS with controlled-release properties. Porous starch, synthesized to overcome the low drug-loading capacity of natural starch, features a high specific surface area and a porous structure [88]. Carbohydrate-based materials derived from natural polysaccharides are biocompatible, biodegradable, and generally recognized as safe, making them suitable excipients for oral drug delivery. In solid SEDDS, they can act as solidification carriers, supporting high drug loading while maintaining self-emulsifying properties.

Porous starch, with its increased surface area and interconnected pore network, effectively adsorbs lipid-based formulations, enabling rapid reconstitution into nanoemulsions upon contact with gastrointestinal fluids and improving the dissolution of poorly water-soluble drugs. Porous starch-based self-assembled nano-delivery systems have been shown to form stable nanocarriers (~30 nm) in gastrointestinal fluids, increasing the aqueous solubility of lipophilic drugs by over 50,000-fold and enhancing oral bioavailability by approximately tenfold compared to free drug suspensions [89,90]. Other carbohydrate-based carriers have also demonstrated strong potential in solid SEDDS design. For example, guar gum–pectin matrices combined with Syloid^®^ have been used for colon-targeted delivery of xanthohumol, enabling microbiota-triggered release with high amorphization and preservation of nanodroplet characteristics [91]. Similarly, lactose-based carriers such as Tablettose^®^ 80, FlowLac^®^ 100, and GranuLac^®^ 200 have been employed to solidify curcumin SEDDS, markedly improving dissolution while showing carrier-dependent differences in powder flow and stability [92]. Maltodextrin has also been applied as a hydrophilic carrier for solid SEDDS, producing a shelf-stable, palatable powder with enhanced oxidative stability, rapid re-dispersion into nanoemulsions, and significantly improved oral bioavailability and pharmacological efficacy [93].

As mentioned earlier, carbohydrate-based materials offer many advantages. However, like polymer-based materials, they also have the disadvantage of requiring large quantities to adsorb and solidify liquid SEDDS. Overall, carbohydrate-based materials can offer versatility and biocompatibility in the design of solid SEDDS.

## 5. Manufacturing Methods for Solid SEDDS

Beyond the selection of solid carriers, the manufacturing method is also a critical factor for the successful development of solid SEDDS (Figure 2). The chosen method significantly influences the drug release rate, redispersion efficiency, and ultimate in vivo drug absorption. Therefore, an optimal solid SEDDS design must consider the manufacturing process alongside the properties of the drug and the solid carrier.

### 5.1. Spherical Crystallization Technology

Spherical crystallization is a particle design technique used to simultaneously improve the flowability and compressibility of drug crystals. This method offers a distinct advantage by addressing drug solubility and stability issues without high-temperature processes, making it suitable for heat-sensitive drugs [94]. The technique utilizes three types of solvents: (1) a good solvent to dissolve the drug, (2) a bad solvent that is immiscible with the good solvent, and (3) a liquid bridging agent to agglomerate the particles. An agitated solution of the drug dissolved in the good solvent is added to the bad solvent. As the good solvent diffuses into the bad solvent, the drug’s solubility rapidly decreases, causing fine drug particles to precipitate. These precipitated particles are then agglomerated into spherical shapes by the liquid bridging agent. Since this entire process occurs simultaneously in a single step, it enables efficient particle production [95]. Notably, this technique does not require a dedicated solid carrier; the drug is precipitated from the good solvent into the immiscible bad solvent and then agglomerated into spherical particles by the liquid bridging agent. Where appropriate, a thin polymer overcoating (e.g., PVP or HPMC) may be applied to enhance mechanical robustness and facilitate downstream handling.

This technology is advantageous due to its simple process, low manufacturing cost, and suitability for heat-sensitive drugs since it avoids high temperatures. It also shows high drug encapsulation efficiency. However, a major concern is the potential for residual organic solvents, which can be harmful to both humans and the environment. Therefore, careful process optimization and management of residual solvents are crucial. Additionally, there can be stability issues, such as an increase in particle size due to the agglomeration of microspheres during storage at high temperatures [96].

Applications of spherical crystallization have shown improvements in powder properties and downstream processability. Reported methods include spherical agglomeration, quasi-emulsion solvent diffusion, crystallo-co-agglomeration, ammonia diffusion, and neutralization. Key process variables such as solvent composition, bridging liquid addition rate, agitation speed, and temperature strongly influence particle size, sphericity, and mechanical strength [97]. In benzoic acid systems, process analytical technology studies have shown that adding the bridging liquid after crystallization yields uniform, highly spherical agglomerates with enhanced flowability and compressibility by optimizing the crystal-to-droplet size ratio [98]. Studies on vanillin and vanillin–salicylic acid have confirmed that selecting an appropriate bridging liquid, controlling the bridging solvent ratio, and adjusting agitation speed can improve sphericity, tabletability, and dissolution performance [99]. When optimized, spherical crystallization methods can be applied to poorly soluble, low-flowability drugs to produce free-flowing particles with excellent compaction properties [100].

### 5.2. Spray Drying

Spray drying is a widely used continuous process in the pharmaceutical industry that transforms a liquid feed into fine solid particles. It is one of the most effective methods for solidifying liquid SEDDS. The spray drying process involves four main steps: atomization, mixing with hot air, evaporation, and product separation [101,102].

First, the liquid SEDDS is dispersed with a solid carrier in a suitable solvent (typically ethanol) to create a spray solution. If the drug is poorly water-soluble, an organic solvent is used, often requiring an inert atmosphere created by nitrogen gas. This solution is then atomized into fine droplets through a spray nozzle. These droplets come into contact with a hot drying gas, and the solvent rapidly evaporates, forming solid particles. The resulting solid powder is then separated from the gas stream and collected using a cyclone separator [103,104]. Spray drying can produce solid powders in a single step, resulting in a uniform particle size distribution (with particles ranging from sub-micron to micron scales). The resulting powder has improved flowability and dissolution rates, and the process is relatively inexpensive and suitable for commercial-scale production [105,106]. However, it is difficult to apply this method to heat-sensitive drugs like proteins or enzymes. Additionally, volatile components of the liquid SEDDS (e.g., surfactants) can be lost during the process, which may affect the final drug loading capacity [107].

Numerous studies have reported the successful application of spray drying for solid SEDDS, resulting in improved powder properties and drug performance. For example, cilostazol-loaded SEDDS spray dried with Aerosil^®^ 200 under controlled conditions produced nanosized amorphous particles with high drug loading and rapid emulsification, leading to markedly improved dissolution compared with the pure drug [108]. In the case of fenofibrate, spray drying with lactose–regulating agent yielded solid SEDDS that retained nanoscale droplet sizes upon reconstitution and enhanced in vivo oral bioavailability [109]. Lipid nanodispersions processed with hydrophilic carriers, particularly lactose–Sodium dodecyl sulfate at optimized inlet temperatures, maintained nanoparticle integrity and produced amorphous powders with excellent redispersibility and flow properties [110]. Recent reviews have emphasized spray drying’s versatility and scalability, noting that appropriate carrier selection and process optimization can ensure robust emulsification performance, improved stability, and compatibility with emerging solidification technologies [20].

### 5.3. Supercritical Fluid (SCF)-Based Methods

SCF technology is a process that uses SCF (e.g., CO_2_), which exhibits properties of both a gas and a liquid under high pressure and temperature. This method has the advantage of minimizing the use of organic solvents and allowing for precise control over particle size and crystal form [111]. The manufacturing process involves dissolving the drug and lipid excipients in an organic solvent (e.g., methanol) and then dissolving this solution in the SCF. As the system’s pressure and temperature are gradually lowered, the solubility of the substance in the SCF decreases. The resulting decrease in solubility leads to the slow infiltration of coating material, forming a layer, or the precipitation of fine drug particles to produce the final product. This process is highly effective for precisely controlling particle size and morphology [112]. While SCF technology offers environmentally safe processing and high precision, it requires expensive equipment and complex process controls to maintain supercritical conditions. Understanding the solubility of formulation components in SCF is also crucial. Despite these challenges, it holds significant potential for optimizing the particle engineering characteristics of solid SEDDS.

Several studies have demonstrated the feasibility of SCF-based methods for solid SEDDS. Techniques such as rapid expansion of supercritical solution, particles from gas-saturated solution, and supercritical anti-solvent can be optimized by adjusting pressure, temperature, solvent type, and nozzle design to fine-tune particle morphology and crystallinity while avoiding residual solvents and heat degradation [113]. Experimental work with supercritical CO_2_ has shown that combining accurate solubility profiling with advanced thermodynamic models. For example, paracetamol crystals were transformed from ~70 μm needles into near-spherical particles ranging from 30 nm to 5 μm, with size distribution controlled by processing parameters [114,115]. These findings suggest that SCF technology offers a solvent-free, scalable approach for producing solid SEDDS with tailored particle characteristics, particularly for drugs with poor solubility or thermal instability.

### 5.4. Adsorption onto a Solid Carrier

Adsorption is one of the most common and economical methods for converting liquid SEDDS into solid SEDDS. This technique involves adsorbing a liquid formulation onto the surface of a porous solid carrier to create a stable, free-flowing powder [22]. Adsorption is a thermodynamically favorable phenomenon where adsorbate molecules (in this case, components of the liquid SEDDS) accumulate on the surface of an adsorbent (the solid carrier). This is an exothermic process that heavily depends on the large surface area and specific surface properties of the adsorbent. The adsorption of liquid SEDDS onto a solid carrier primarily occurs through physisorption, which involves weak forces such as Van der Waals forces or electrostatic interactions. This allows for the formation of multiple layers on the adsorbent’s surface [116]. This method is highly economical as it can be easily manufactured using simple equipment like a mortar and pestle or a mixer, without the need for complex machinery. It produces a free-flowing, stable solid powder that is easy to handle and store. However, a large amount of solid carriers are required to sufficiently adsorb the liquid formulation, which can increase the final volume of the dosage form. When the adsorbed liquid components are compressed into tablets, the liquid phase can exude, leading to problems such as softening, tackiness, chipping, or changes in hardness. Finally, if the lipid components penetrate deep into the carrier’s pores or aggregate over time, emulsification and hydration can become difficult, resulting in a decrease in drug release [117,118].

Several studies have reported the successful application of this method to produce solid SEDDS with improved dissolution, stability, and bioavailability. Olmesartan medoxomil-loaded SEDDS was solidified using Aerosil^®^ 200 and Avicel^®^ PH101, producing nanosized amorphous particles with rapid emulsification and markedly enhanced dissolution compared with the pure drug [119]. Similarly, tenofovir-loaded liquid SEDDS was adsorbed onto carriers such as Neusilin^®^ US2, mannitol, and calcium silicate by simple blending, resulting in free-flowing powders [120]. Silica carrier studies have shown that larger pore sizes can enhance dissolution efficiency and wettability, particularly when pretreated with polyvinylpyrrolidone [121]. Adsorption of a tocotrienol-rich fraction onto magnesium aluminosilicate produced stable emulsions with droplet sizes of 210–277 nm, resulting in a 3.4–3.8-fold increase in in vivo oral bioavailability in rats [122].

In conclusion, adsorption onto a solid carrier is one of the most practical methods for manufacturing solid SEDDS. However, to address its disadvantages, such as the large volume of the formulation and the potential for inhibited drug release, further research and technological development are needed. This includes modifying the carrier surface and using optimal compression excipients.

### 5.5. Hot Melt Extrusion (HME)

HME is a manufacturing process that creates solid, uniform products by melting or softening materials under high temperature and pressure, then forcing them through a die. The HME process uses an extruder composed of an extrusion barrel, a rotating screw (single or twin), a motor, heaters, and a die. The polymer matrix (binder) is melted or softened by both the frictional heat generated by the rotating screw inside the barrel and external heaters. For solid SEDDS, the feed strategy can be configured in a single step using zonal feeding: the drug and lipids are introduced into the first feeding zone and, upon heating, form a molten SEDDS phase; the solid carrier is then added via a downstream (side) feeder to yield a uniform solid/semi-solid extrudate. Alternatively, split feeding introduces the liquid SEDDS and the solid carrier in different barrel zones to control dispersion and minimize thermal/mechanical stress [123,124]. Depending on the die configuration, the molten material can be shaped into various products: when the die is attached to the barrel, films, filaments, or pellets (by cutting filaments into pellet size) can be produced, whereas detaching the die allows direct extrusion of granules. Because lipids are often included in SEDDS formulations, the processing temperature can be lowered below the melting point of the drug or the glass transition temperature of the polymeric carrier, thereby reducing the risk of thermal degradation. HME offers several advantages: it is a solvent-free process, which eliminates safety concerns related to residual solvents. It is also a continuous and efficient process with a short production time. Furthermore, it enhances drug content uniformity and can create various dosage forms (e.g., films, granules). However, there are disadvantages: the use of high temperature and pressure poses a risk of degradation for heat-sensitive drugs. The process also requires a significant amount of energy. Additionally, the high shear stress can lead to the mechanical degradation of the polymer matrix [125].

Several studies have demonstrated the potential of HME for manufacturing solid SEDDS, enabling solvent-free, continuous processing with precise control over drug dispersion and release. For example, fenofibrate-loaded solid SEDDS was prepared using a single-step, continuous twin-screw HME process with Neusilin^®^ US2 as the solid carrier, yielding amorphous granules with rapid drug release (>90% in 15 min) and stable performance over three months. The study also emphasized scalability, process automation, and flexibility in incorporating various drugs and carrier systems [124,126]. In another pilot-scale feasibility study, a twin-screw extruder achieved up to 30% SEDDS loading through split feeding, producing amorphous extrudates that rapidly reconstituted into nanoemulsions with droplet sizes of 50–300 nm. Complementary analysis highlighted the importance of quality by design principles, process analytical technology, and tailored screw configurations for robust scale-up [125,127].

HME presents great potential for manufacturing solid SEDDS by offering a solvent-free, high-efficiency process. Its ability to precisely control drug release, especially when combined with pH-sensitive matrices, is a significant advantage. However, its limitation in handling thermally unstable drugs necessitates careful formulation design that considers the drug’s stability.

### 5.6. Pellet Manufacturing Technology

Pellets are spherical particles with a uniform size distribution, which can be produced using the extrusion–spheronization method. This method involves adding a binder that melts or softens at high temperatures to a powdered material, causing the powder to agglomerate [128,129]. Semi-solid lipids can be used as a binder. This process differs from wet granulation in that it does not require a drying step [130].

The wet mass, mixed with the molten binder, is passed through an extruder (e.g., a screw extruder) to form rod-shaped extrudates. The size of the extruder die opening determines the final pellet size. The extruded rods are then placed in a spheronizer and rotated. The friction forces shape them into uniform, spherical pellets. The quality of the final pellets is influenced by various parameters, including the mixer rotation speed, the particle size, concentration, and viscosity of the meltable binder. Semi-solid self-emulsifying lipids can be integrated with inert solid carriers like silica to form pellets. These pellets have excellent flowability due to their spherical shape, a narrow size distribution, and allow for easy, uniform coating and dense packing [131].

Several studies have shown that process optimization can significantly enhance pellet properties for solid SEDDS applications. For example, systematic variation of extrusion and spheronization parameters, including die configuration and spheronizer speed, produced highly spherical κ-carrageenan-based pellets (aspect ratio < 1.1) with narrow size distribution, mechanical strength, and consistent dissolution across different APIs [132]. In solid lipid-based pellets, precise thermal control during spheronization using infrared heating and temperature monitoring allowed continuous processing at high lipid loads, resulting in pellets with consistent sphericity and stable lipid polymorphism [133]. Evaluations of self-emulsifying granules and pellets have shown that emulsion viscosity, surfactant hydrophilicity, and spheronization time significantly affect pellet roundness, density, and reconstitution performance, providing a predictive framework for quality optimization [130,134]. Furthermore, carvedilol-loaded self-emulsifying pellets developed via extrusion–spheronization demonstrated that selecting appropriate carriers, disintegrants, and process conditions can yield porous, spherical pellets capable of rapid reconstitution into nanoemulsions, achieving over 90% drug release within 30 min and consistent dissolution across physiological pH ranges [135]. Specifically, pellets produced via the extrusion-spheronization method offer a solvent-free process, which increases efficiency, and provides the flexibility to control drug release characteristics through various formulation parameters.

### 5.7. Manufacturing Solid SEDDS Using Hard Gelatin Capsules

Using hard gelatin capsules is the simplest method for converting liquid SEDDS into a solid dosage form. This approach addresses the issue of low patient compliance with liquid drug formulations and offers the advantages of easy handling and storage [136]. In practice, liquid SEDDS are often filled into soft gelatin capsules (considered solid oral dosage forms), whereas hard gelatin capsules are more commonly used when the formulation is adsorbed onto a solid carrier or contains solid/semi-solid lipids. This method involves directly filling hard gelatin capsules with the liquid SEDDS. The appropriate capsule size is selected based on the drug dose of the final formulation. A critical aspect of this method is the careful evaluation of compatibility, ensuring that the liquid SEDDS components (oils, surfactants, co-surfactants, etc.) do not chemically react with or physically affect the capsule shell [137]. While conceptually straightforward, direct liquid filling into hard gelatin capsules generally requires specialized liquid-filling equipment and capsule band sealing, and physical stability of the shell (softening, deformation, leakage) should be carefully monitored. However, when liquid SEDDS are handled in hard gelatin capsules, the need for specialized equipment and sealing can offset the economic advantage. Nevertheless, there are potential drawbacks. The liquid SEDDS can soften or dissolve the capsule shell, leading to leakage issues. To prevent this, additional processes such as capsule band sealing may be necessary. Furthermore, the fixed volume of the capsule limits the amount of liquid SEDDS that can be filled, which can be a significant disadvantage for high-dose drugs [138].

Several studies have demonstrated the practicality of this method. In one example, candesartan cilexetil SEDDS optimized for droplet size (<50 nm) was filled into size No. 2 capsules using semi-automatic equipment, followed by gelatin band sealing to prevent leakage [139]. In another study, lovastatin SEDDS was adsorbed onto spray-dried magnesium aluminometasilicate before being filled into hard gelatin capsules, with control of the liquid-to-powder ratio ensuring high drug loading, flowability, and capsule integrity [140]. From a manufacturing perspective, successful liquid filling depends on controlling fill volume (typically ≤ 90% of capsule capacity), viscosity (0.1–1 Pa·s), and temperature to prevent shell deformation, along with sealing techniques such as gelatin banding or hydro-alcoholic fusion to avoid leakage [141]. More recently, carvedilol SEDDS with optimized oil/surfactant ratios were filled into size No. 3 capsules, producing stable formulations with rapid drug release and preserved droplet size after dispersion, confirming the method’s reproducibility and scalability [142].

### 5.8. Lyophilization (Freeze-Drying)

Lyophilization is a drying technique widely used in the pharmaceutical industry to enhance the stability of drugs significantly. This method can convert liquid SEDDS into a solid powder with minimal moisture content [143]. The manufacturing process is as follows: First, the liquid SEDDS is completely frozen to a solid state. During this step, it is crucial to use a cryoprotectant to ensure the stability of the drug and the formulation. The frozen formulation is then placed under high vacuum, and the temperature is slowly raised. The frozen solvent undergoes sublimation, transitioning directly to a gaseous state without passing through a liquid phase. This process removes 95–99.5% of the water. After the primary drying, a small amount of bound moisture may remain in the formulation. To remove this residual moisture, the temperature is raised further to promote desorption. The final dried product is then sealed under an inert gas or vacuum to prevent oxidation [144].

This method offers several advantages: the moisture content is reduced by over 95%, which significantly improves long-term storage stability. It is particularly effective for stabilizing substances that are susceptible to oxidation. The low-temperature process also prevents the degradation of heat-sensitive drugs. Lyophilized solid SEDDS generally exhibit excellent stability and flowability [145]. However, there are also disadvantages: the process is very expensive due to the need for equipment that maintains high vacuum and low temperatures. Additionally, volatile compounds (e.g., ethanol, isopropanol, acetone, ethyl acetate) can be removed under high vacuum conditions, which may affect the final composition of the formulation [146].

Several studies have successfully applied lyophilization to convert liquid SEDDS into stable, free-flowing powders with improved solubility and dissolution. Fenofibrate-loaded liquid SEDDS frozen with trehalose as a cryoprotectant and lyophilized for 72 h produced amorphous powders with increased solubility, excellent flowability, and immediate-release tablet performance [147]. Papain-loaded SEDDS lyophilized in the presence of trehalose yielded porous powders that retained enzymatic activity, showed excellent re-dispersibility, and maintained long-term stability [148]. Telmisartan–polymer dispersions processed under optimized primary and secondary drying conditions produced amorphous solid dispersions with improved dissolution and bioavailability [14]. A novel in situ freeze-drying approach performed directly within hard gelatin capsules generated amorphous nifedipine–polyvinylpyrrolidone powders with enhanced dissolution while preserving capsule integrity [149].

### 5.9. Three-Dimensional Printing (Three-Dimensional Printing)

3D printing technology has recently garnered attention as an innovative approach in pharmaceutical manufacturing. This technology provides the flexibility to precisely fabricate patient-specific drug delivery systems, opening up new possibilities to overcome the limitations of conventional manufacturing methods [150,151,152]. Prominent 3D printing techniques used for manufacturing solid SEDDS include Fused Deposition Modeling (FDM) and semisolid extrusion. In FDM, a filament containing the solid SEDDS is melted and precisely extruded through a nozzle, building up the desired tablet shape layer by layer. The semisolid extrusion method uses paste-like material containing solid SEDDS, which is directly deposited to form the desired structure [153]. Three-Dimensional printing allows for precise control over drug release profiles. For example, by optimizing the tablet wall thickness, nozzle size, and printing parameters, the drug release rate can be controlled. Formulations can also be designed to degrade only under specific conditions within the gastrointestinal tract (e.g., pH changes). Furthermore, the dosage, shape, and release characteristics can be customized to individual patient needs. Combining lipid-based carriers with a 3D-printed matrix allows for the creation of formulations that contain multiple drugs in a single tablet or have complex internal structures [154].

Several studies have demonstrated the applicability of 3D printing for solid SEDDS with precise geometry and tailored release characteristics. Solid SEDDS of fenofibrate-loaded lipid-based emulsion gels at room temperature produced tablets with high mass uniformity (<5% variance), rapid disintegration (<15 min), and reproducible self-emulsification without compromising thermolabile drug stability [155]. Reviews of extrusion-based 3D printing for nanomedicine-loaded solids have emphasized the optimization of rheological properties, nozzle dimensions, and extrusion pressure to maintain print fidelity, while also noting the limitations of heat-intensive FDM for thermosensitive systems [156]. Solid SEDDS printing of dapagliflozin-containing SEDDS pastes prepared by fusing lipid–surfactant matrices with drug-loaded oils enabled dose personalization through controlled tablet sizing, while preserving droplet size (~105 nm) and emulsification performance after dispersion [157]. In another example, polymer–lipid-hybrid scaffolds fabricated by FDM and subsequently filled with solid SEDDS created multi-compartment dosage forms with controllable surface area-to-volume ratios and the ability to incorporate multiple drugs into a single tablet [158]. Shape-engineered printing of Gelucire^®^-based solid SEDDS without additional excipients demonstrated that increasing surface area-to-volume, as in torus geometries, significantly accelerated dispersion and lipid digestion rates in vitro [159]. Three-Dimensional printing is a technology linked to personalized medicine, going beyond just improving the stability and patient compliance of solid SEDDS formulations. However, challenges such as manufacturing speed and cost still need to be addressed [160].

Table 1 provides a comprehensive comparison of the key features, advantages, and disadvantages of all the solid SEDDS manufacturing methods discussed so far.

## 6. Applications of Solid SEDDS

### 6.1. Enhanced Bioavailability and Protein Delivery

The most fundamental role of SEDDS is to enhance the absorption of hydrophobic drugs by forming a nano- or micro-sized oil-in-water emulsion [161,162]. Solid SEDDS effectively perform this function while overcoming the drawbacks of conventional liquid SEDDS (e.g., toxicity, instability). Solid SEDDS can further maximize bioavailability by using polymeric precipitation inhibitors to prevent drug crystallization and maintain the drug in a prolonged supersaturated state. This also offers the benefit of reducing the use of excessive surfactants [163,164,165,166].

SEDDS can also be applied to the delivery of proteins and peptides, which are difficult to administer orally due to their high hydrophilicity and low stability [167]. The strategy of ion-pair formation to increase the lipophilicity of proteins and reduce leakage is a crucial solution in this field. By protecting peptides from the harsh gastrointestinal environment and bypassing enzymatic degradation, this approach enables the oral administration of biologics, which can reduce reliance on injections and significantly improve patients’ quality of life [168].

Recent studies have demonstrated the feasibility of solid SEDDS for oral protein and peptide delivery. Lysozyme complexed with sodium dodecyl sulfate and adsorbed onto solid carriers retained bioactivity and self-emulsification properties, producing a stable powder suitable for oral administration [169]. Insulin ion-paired with sodium n-octadecyl sulfate showed enhanced lipophilicity and was incorporated into an SEDDS, providing enzymatic protection, improved intestinal permeability, and enhanced hypoglycemic effects in diabetic rats [170]. Exenatide complexed with sodium n-octadecyl sulfate and other counterions achieved significantly increased intestinal membrane permeability and up to 19.6% relative oral bioavailability compared with subcutaneous injection when delivered via SEDDS [171]. These findings highlight the synergistic effect of ion pairing and solid SEDDS in enhancing stability, permeability, and bioavailability for oral peptide formulations.

### 6.2. Controlled Drug Release

Solid SEDDS can be developed as sustained-release formulations to precisely control the drug dissolution profile, prolonging drug efficacy and minimizing fluctuations in plasma concentration. This is particularly beneficial for drugs with a short half-life or a narrow therapeutic index. Sustained-release properties can be conferred upon solid SEDDS through various techniques, including microencapsulation, pelletization, polymer coating, and matrix-based tablets [172,173].

Osmotic pump technology has been successfully combined with solid SEDDS to provide precise, zero-order release while maintaining self-emulsifying performance. For example, a solid self-emulsifying osmotic pump tablet of nifedipine was prepared by adsorbing a Gelucire^®^/Lutrol^®^/Transcutol^®^-based formulation onto inert carriers, followed by a semipermeable membrane coating and laser-drilling an orifice. The system maintained droplet size after reconstitution and achieved approximately 84% cumulative release over 12 h, independent of agitation speed, with the release rate controlled by membrane thickness, plasticizer content, and orifice diameter [174]. Similarly, cyclosporine A-loaded SEDDS were incorporated into osmotic pump tablets using cellulose acetate membranes. Optimization of osmotic agent ratios, pore-former levels, and orifice size enabled sustained nanoemulsion release for 12 h. In vivo studies in dogs showed prolonged Tmax, extended mean residence time, and reduced Cmax compared with immediate-release formulations, confirming the clinical potential of combining SEDDS with osmotic delivery for controlled absorption of poorly soluble drugs [175]. The design of osmotic SEDDS is also possible. By using osmotic agents like mannitol, water is allowed to enter the system through a semipermeable membrane. This pressure is then leveraged to release the drug at a precisely controlled rate.

Furthermore, floating and gastroretentive SEDDS can be designed. These formulations are engineered to remain in the stomach for an extended period, which optimizes the drug absorption window and prevents drug instability in alkaline environments [173].

Floating and gastroretentive S-SEDDS formulations are designed to achieve immediate buoyancy and controlled drug release within the gastric environment. By incorporating low-density excipients, such as functionalized calcium carbonate, and effervescent agents, like sodium bicarbonate, these systems can maintain self-emulsifying properties while achieving prolonged buoyancy [176]. For example, SEDDS containing fenofibrate demonstrated sustained-release effects of more than 12 h and enhanced therapeutic efficacy, demonstrating the feasibility of combining self-emulsifying and buoyancy technologies [173]. Gastroretentive tablets containing hydrophilic polymers and gas-generating agents demonstrated rapid buoyancy and sustained-release kinetics, indicating the clinical potential of this approach. Incorporating poorly soluble curcumin into S-SEDDS increased solubility and permeability, which subsequently facilitated gastric buoyancy behavior and prolonged retention [177]. Collectively, these results demonstrate that the buoyant S-SEDDS platform can effectively enhance bioavailability through prolonged gastric retention and sustained release.

### 6.3. Mucoadhesive Technology

Mucoadhesive polymers, such as chitosan and thiolated polyacrylic acid derivatives, have been extensively studied for their ability to adhere to mucosal surfaces, enhancing drug bioavailability and thereby extending drug retention at the site of absorption [178,179]. SEDDS, known to improve the solubility and oral bioavailability of poorly soluble drugs, can be effectively combined with mucoadhesive polymers to further optimize drug delivery. This combination leverages the advantages of SMEDDS in drug solubility and the prolonged mucosal interaction provided by the mucoadhesive polymer, enhancing local delivery and systemic absorption. For example, mucoadhesive SMEDDS containing thiolated chitosan have been shown to adhere more strongly to the gastrointestinal mucosa via covalent disulfide bonds with mucus, resulting in prolonged drug retention and an improved pharmacokinetic profile [180]. Furthermore, multifunctional designs, such as nanofibrous SEDDS incorporating a thiolated polyacrylic acid mucoadhesive layer and SEDDS, have demonstrated enhanced mucoadhesion and drug absorption [181]. This innovative mucoadhesive SMEDDS approach offers a promising strategy to address the challenges of oral drug delivery by contributing to sustained drug release, enhanced bioavailability, and targeted drug delivery.

### 6.4. Targeted Drug Delivery

Solid SEDDS can enable the selective delivery of drugs by incorporating ligands that target specific tissues or cells. This approach not only improves therapeutic efficacy by increasing local drug concentration at the intended site but also reduces systemic side effects through minimized off-target distribution [179]. Functionalization strategies include conjugation of peptides, antibodies, aptamers, or small molecules to the lipid-based carrier surface, enabling receptor-mediated uptake or tissue-specific accumulation [180]. Furthermore, the inherent physicochemical versatility of solid SEDDS facilitates the integration of multiple functionalities, such as controlled or stimuli-responsive drug release (e.g., pH, enzyme, or temperature triggers), and co-delivery of synergistic agents [181]. These capabilities position solid SEDDS as a promising platform for precision medicine, with applications in oncology, metabolic disorders, and the oral delivery of biopharmaceuticals [182].

Recent advances have demonstrated the feasibility of incorporating targeting moieties into solid SEDDS to achieve site-specific delivery for both small molecules and biologics. For example, lysozyme was ion-paired with sodium dodecyl sulfate to enhance lipophilicity and subsequently incorporated into a lipid-based carrier, preserving enzymatic activity while improving gastrointestinal stability and residence time. The formulation, adsorbed onto porous carriers, retained self-emulsifying ability and offered potential for targeted intestinal absorption [169]. In oncology, tumor-homing peptides have been integrated into self-microemulsifying systems to direct therapeutic payloads to breast cancer cells via receptor-mediated endocytosis, producing stable nanoemulsions (~20 nm) with selective cellular internalization [183]. Additionally, modular SEDDS have been designed to co-deliver cisplatin and siRNA using water-in-oil-in-water emulsions, optimizing droplet size and surface charge to enhance uptake by triple-negative breast cancer cells while reducing off-target toxicity [184]. In metabolic disease applications, ligand-functionalized lipid nanoparticles incorporating polymer–lipid hybrids have been engineered for adipose tissue targeting, with stimuli-triggered release mechanisms that improve site-specific drug accumulation [44]. Collectively, these strategies demonstrate how surface functionalization, ion-pairing, and modular emulsion architectures can expand the utility of solid SEDDS for precision therapeutics, combining high delivery efficiency and targeting specificity with scalable manufacturing and formulation stability.

### 6.5. Personalized Medicine

As previously discussed in Section 5.9, Solid SEDDS enables personalized medicine through integration with 3D printing technology. This allows for the precise optimization of a drug’s release profile, dosage, and shape to match the individual physiological characteristics of a patient. This approach can enhance patient compliance in the treatment of complex chronic diseases and will significantly contribute to addressing inter-individual variability in drug metabolism and response [185]. The applications of the aforementioned solid SEDDS are summarized in Table 2.

## 7. Challenges and Limitations of Solid SEDDS

While solid SEDDS are a powerful solution for enhancing the bioavailability of poorly water-soluble drugs, several technical, regulatory, and commercial challenges must be overcome for widespread application. Despite significant research in the solid SEDDS field, there is still a lack of successfully commercialized products due to a few critical barriers.

### 7.1. Low Drug Loading

One of the biggest commercial hurdles for solid SEDDS is low drug loading. Liquid SEDDS already have a solubility limit for poorly soluble drugs, and the addition of a solid carrier further dilutes the drug concentration within the formulation. This makes it challenging to apply solid SEDDS to high-dose drugs or those with low solubility in lipid excipients. A promising approach to overcome this problem is to adsorb SEDDS onto carriers with high surface area and high surface porosity, thereby increasing drug loading. This approach offers the potential to maximize the final drug content in solid SEDDS.

### 7.2. Stability and Compatibility Issues

While solid SEDDS address the limitations of liquid SEDDS, they still have their unique stability problems. The solidification techniques may still lead to a degradation of the formulation’s physicochemical stability in hot and humid environments. Although solid SEDDS show excellent efficacy for hydrophobic drugs, their application to hydrophilic or macromolecular drugs remains very limited. New approaches, such as ionic liquids or hybrid delivery systems, are being proposed as alternatives, but they require further research.

### 7.3. Key Commercial Considerations

Solid Self-Emulsifying Drug Delivery System (SEDDS) technology has yielded products that have either received FDA approval or are currently undergoing clinical trials, as outlined in Table 3. To further enhance the market potential of solid SEDDS products, the following commercial considerations must be satisfied: (1) Cost-effectiveness and scalability: The manufacturing process must be cost-effective and easily scalable for mass production. (2) Biocompatibility: All excipients used must ensure long-term safety. (3) Standardization: There is a need for standardized process development to control various variables and produce reproducible solid SEDDS. (4) Solidification and stabilization: To improve the production and stability of solid SEDDS, a comprehensive review of the use of solid carriers (diluents/polymers), stabilizers, and antioxidants is required for successful commercialization.

## 8. Conclusions and Perspectives

Liquid SEDDS have led to innovation by solving one of the biggest challenges in the pharmaceutical industry: the solubility and bioavailability of poorly water-soluble drugs. Despite their excellent bioavailability-enhancing effects, liquid SEDDS have faced commercialization difficulties due to stability issues during storage (phase separation, drug precipitation) and gelatin capsule leakage. The emergence of solid SEDDS marked a critical turning point in overcoming these limitations. With the introduction of various advanced technologies like spray drying, adsorption, hot melt extrusion, and 3D printing, it has become possible to create solid formulations that maintain the nanostructure of liquid SEDDS while maximizing stability and portability. In particular, the solid carrier is accelerating progress in this field. Previously considered a passive excipient that simply adsorbed liquid lipids, recent studies have demonstrated that solid carriers are key variables that can actively control drug release rates, stability, and even in vivo performance.

The value of solid SEDDS extends beyond simply increasing the absorption rate of hydrophobic drugs. Through integration with advanced technologies, SEDDS are now evolving into multifunctional platforms with expanded capabilities tailored to specific therapeutic goals. While solid SEDDS hold immense potential, challenges remain. Challenges such as the toxicity of high-concentration surfactants, formulation limitations for drugs with low lipid solubility, liquid loading capacity for solid carriers, and the technical complexity of manufacturing scale-up require continuous research and innovation. However, these challenges also represent opportunities for greater development. With up to 70% of new drug candidates worldwide exhibiting low aqueous solubility, solid SEDDS remain one of the most effective solutions. Advanced research into hybrid systems and the delivery of biologics will continue to push the boundaries of SEDDS in the future.

## Figures and Tables

**Figure 1 pharmaceuticals-18-01550-f001:**
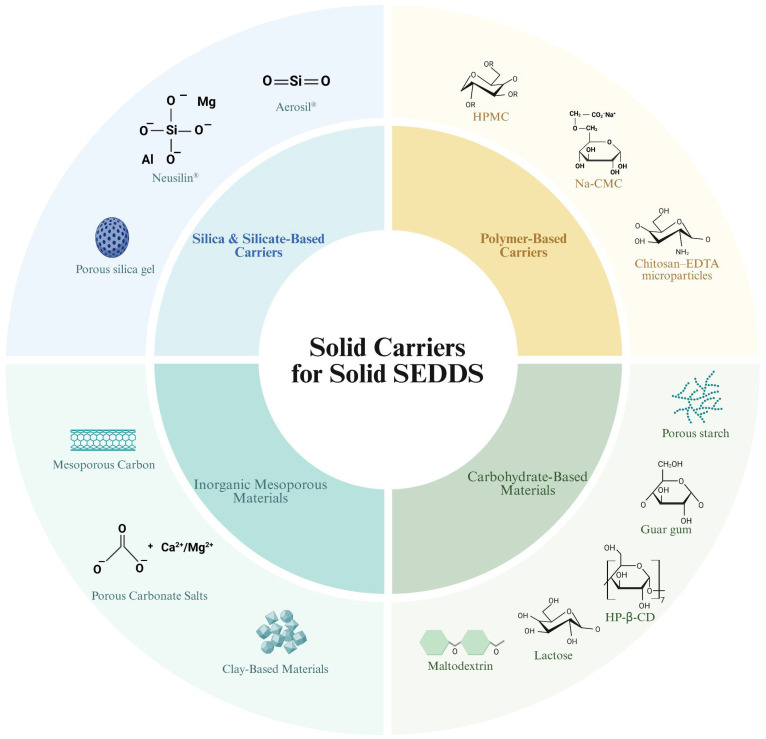
Classification of solid carriers for Solid SEDDS with representative examples.

**Figure 2 pharmaceuticals-18-01550-f002:**
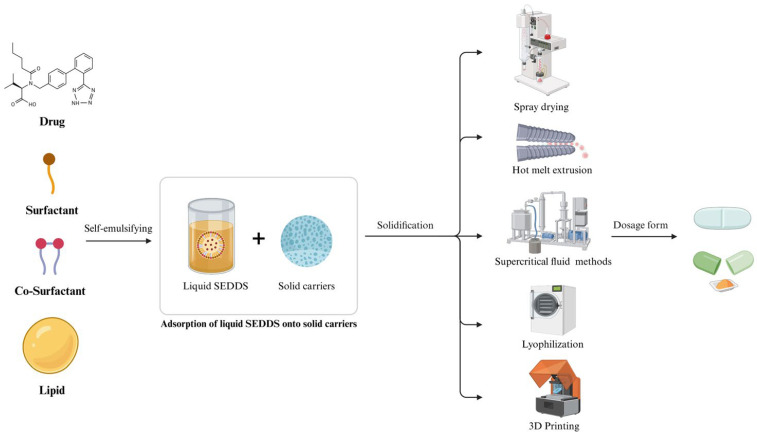
Schematic overview of Liquid SEDDS solidification via solid carriers and representative manufacturing methods for Solid SEDDS.

**Table 1 pharmaceuticals-18-01550-t001:** Solid SEDDS manufacturing methods.

Manufacturing Method	Key Features	Advantages	Disadvantages	Materials	Critical Process Parameters
Spherical Crystallization	Single-step process forming spherical crystals from solvents and a bridging agent	Simple and cheap process, no high temperature, good flowability, and compressibility.	Use of organic solvents (safety concerns), potential instability at high storage temperatures.	Good solvent, poor solvent, bridging liquid; optional PVP/HPMC.	Solvent ratio, bridging-liquid addition rate, agitation speed, temperature.
Spray Drying	Liquid feed (SEDDS + carrier) is atomized and dried with hot gas.	Single-step continuous process, uniform particle size, improved dissolution rate, suitable for scale-up.	High-temperature process (unsuitable for heat-sensitive drugs), potential loss of volatile components, limited to certain carriers.	Hydrophilic carriers: Aerosil^®^ 200, lactose(+SDS), mannitol, Ca-silicate; optional PVP.	Atomization/nozzle, inlet–outlet temperature, gas flow, feed solvent.
Supercritical Fluid (SCF)	Solubility change in SCF	Solvent-free, precise particle control	Expensive equipment, complex process	SC-CO_2_; drug/lipids in organic solvent.	Pressure, temperature, solvent choice, nozzle/expansion
Adsorption onto Solid Carrier	Liquid SEDDS absorbed by a porous solid carrier to form a free-flowing powder.	Economical, simple process (no complex equipment), improved stability and flowability.	Requires large carrier volume, risk of liquid exudation during compression, and potential for reduced drug release efficiency	Aerosil^®^ 200, Neusilin^®^ US2, Avicel^®^ PH101, mannitol, Ca-silicate; PVP-treated silica optional.	Liquid/solid ratio, pore size, surface chemistry, mixing time/speed.
Hot Melt Extrusion (HME)	SEDDS components are blended and extruded through a die at high temperature and pressure.	Solvent-free, continuous, and efficient, enhanced drug uniformity, versatile shapes (strands, films).	High temperature/pressure risk, drug degradation, high energy consumption, and high shear stress can degrade the polymer matrix.	Solid carriers incl. Neusilin^®^ US2; optional pH-responsive polymers.	Barrel T profile, screw design/speed, feed strategy (split feeding), die.
Pelletization	Liquid SEDDS incorporated into a binder, extruded, and spheronized.	Excellent flowability, narrow size distribution, easy to coat, efficient solvent-free process.	Complex process control (multiple variables) requires specific equipment, not suitable for all materials.	κ-Carrageenan; semi-solid lipids (Gelucire^®^, glyceryl behenate); silica/MCC.	Die size, spheronizer speed/time, binder viscosity/temperature.
Hard Gelatin Capsule Filling	Liquid SEDDS filled directly into hard gelatin capsules.	Simple, economical, no special equipment, addresses patient compliance issues with liquid dosage forms.	Risk of capsule leakage, limited drug loading capacity, and potential chemical/physical interactions with the capsule shell.	Liquid SEDDS (e.g., <50 nm on dispersion); optional spray-dried Mg aluminometasilicate; band sealing.	Fill volume ≦ ~90%, viscosity ~0.1–1 Pa·s, fill temperature, sealing.
Lyophilization	Liquid SEDDS frozen and dried by sublimation under vacuum.	Low-temperature process (suitable for heat-sensitive drugs), high stability (low moisture content), excellent redispersion.	High cost (equipment and energy), potential loss of volatile components under vacuum, and a time-consuming process.	Cryoprotectants (trehalose); PVP/HPMC matrices; in-capsule option.	Freezing protocol, shelf T ramp, chamber pressure (vacuum), primary/secondary drying time.
3D Printing	Layer-by-layer fabrication using FDM or semi-solid extrusion.	Precise control over drug release, personalized medicine potential, complex dosage forms, multiple drug loading.	Slow manufacturing speed, high cost, limited choice of materials, and regulatory challenges.	Gelucire^®^-based solid SEDDS; polymer–lipid hybrids (PVA/PLA, poloxamer).	Nozzle diameter, infill %, wall thickness, extrusion speed/pressure; T control for FDM.

**Table 2 pharmaceuticals-18-01550-t002:** Application of Solid SEDDS.

Category	Target/ Drug	Key Strategy	Dosage Form	Key Excipients	Key Outcomes	Refs.
Enhanced Bioavailability & Protein Delivery	Poorly water-soluble drugs	Self-emulsifying micro/nano-structuring	Solid SEDDS	PVP, SLS, Copovidone, Labrasol, Peceol, Mesoporous silica, HP-β-CD, etc.	Crystallization suppressed; supersaturation sustained; lower surfactant requirement	[161,162,163,164,165,166]
	Lysozyme	SDS ion-pairing → lipophilicity ↑, leakage ↓	Solid powder	Neusilin^®^ UFL2, Syloid^®^ 244 FP	Enzymatic activity retained; suitable for oral dosing	[169]
	Insulin	Ion-pairing with sodium n-octadecyl sulfate	Solidified after SEDDS loading	Capmul MCM, Labrasol, Tetraglycol	Enzymatic protection; intestinal permeability increased; stronger hypoglycemic effect (rat)	[170]
	Exenatide	Ion-pairing with long-chain anions	SEDDS-based oral delivery	Capmul MCM EP, Captex 355, Kolliphor RH40, sodium n-octadecyl sulfate	Relative oral bioavailability up to 19.6% vs. SC	[171]
Controlled Drug Release	Nifedipine	SEDDS adsorption + semipermeable coating; laser-drilled orifice	Osmotic pump tablet	Gelucire^®^/Lutrol^®^/Transcutol^®^, Aerosil 200, etc.	~84% released over 12 h; reconstituted droplet size preserved; release insensitive to agitation	[174]
	Cyclosporine A	Cellulose acetate osmotic membrane; optimized osmotic/pore-former levels	Osmotic pump tablet	Labrafil M 1944CS, Cremophor EL, Polyethylene oxide, etc.	Sustained 12 h release; Tmax, MRT increased; Cmax reduced (dog)	[175]
	Ginkgolides	Swellable, gas-generating matrix	Floating tablet	HPMC 4KM/E15LV, NaHCO_3_, etc.	Floating lag < 1.5 s; total floating > 12 h; Zero-order release	[176]
	Fenofibrate	Swellable matrix	Gastro retentive tablet	Metolose^®^ 90SH-100000SR, etc.	12 h extended release	[173]
	Curcumin	gas-generating; improved bioavailability	Floating SEDDS powder	Sodium alginate, HPMC K100M, NaHCO_3_, etc.	20-fold increase in anti-oxidant and 10-fold increase in anti-inflammatory activities	[177]
Mucoadhesive Technology	Azithromycin	Covalent disulfide bonding with mucin	Gastro retentive SEDDS	Thiolated pluronic	72 h extended release	[180]
	Buccal mucosa	Mucoadhesive fiber	Patch	polyacrylic acid thiomer	~200× buccal adhesion; sustained release 4 h	[181]
Targeted Drug Delivery	Lysozyme	SDS ion-pairing + lipid carrier	Solid SEDDS (adsorbed)	Miglyol 812, Sodium lauryl sulfate, Tween 80, etc.	GI stability and residence increased; self-emulsification maintained (targeted uptake potential)	[169]
	LyP-1	Tumor-homing peptide-decorated SMEDDS	Nanoemulsion	Peceol, Labrasol, PEG 300	~20 nm droplets; receptor-mediated selective uptake increased	[186]
	Cisplatin + siRNA	W/O/W modular architecture; surface-charge optimization	Modular SEDDS	Labrafac PG, Labrasol, Gelucire 4414, Compritol 888 ATO	Uptake in TNBC cells increased; toxicity reduced; synergistic combination effect	[187]
Personalized Medicine	Various APIs	3D printing for precise control of dose/release/geometry	FDM/semisolid-extruded S-SEDDS tablets	PVA, HPMC, PEG, Capryol 90, etc.	Personalized dosing and release; improved treatment adherence	[188]

**Table 3 pharmaceuticals-18-01550-t003:** Solid SEDDS products approved by the FDA or undergoing clinical trials.

Brand Name/ Phase	Drug	Company	Dosage Form	Key Excipients	Stabilizer	Refs.
Neoral^®^	Cyclosporine A	Novartis	Soft gelatin capsule	Cremophor RH40, propylene glycol, dehydrated alcohol, corn oil mono/di-glycerides	DL-α-tocopherol	[189,190]
Norvir^®^	Ritonavir	AbbVie	Soft gelatin capsule	Cremophor EL, oleic acid	BHT	[191,192]
Fortovase^®^	Saquinavir	Roche	Soft gelatin capsule	Medium-chain mono/di-glycerides, povidone K30	DL-α-tocopherol	[193,194]
Agenerase^®^	Amprenavir	Glaxo Wellcome/ GSK	Soft gelatin capsule	PEG 400, propylene glycol	TPGS	[21,195]
Targretin^®^	Bexarotene	Ligand	Soft gelatin capsule	PEG 400, polysorbate 20, povidone	BHT	[196,197]
Lipofen^®^	Fenofibrate	Cipher	Hard gelatin capsule	Gelucire^®^ 44/14, PEG 20,000, PEG 8000, sodium starch glycolate	Hydroxypropylcellulose	[198,199]
Phase 1	Rencofilstat (CRV431)	Hepion Pharmaceuticals	capsule	Polysorbate 80, PEG-400)	Medium Chain Triglycerides	[200]
Phase 1	Icosabutate (NST-4016)	NorthSea Therapeutics BV	capsule	Not specified	Not specified	[201]

## Data Availability

The data presented in the study are included in the article and available on request from the corresponding author.

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
