# Peer review of "Solidification Materials and Technology for Solid Self-Emulsifying Drug Delivery Systems"

_pharmaceuticals, 2025, doi:10.3390/ph18101550_

Round 1

Reviewer 1 Report

Comments and Suggestions for Authors

Dear Authors,

Thanks for your contribution. You have compiled a significant information on solidification techniques and applications, however i would like to suggest you for further improvement. Applications of Solid SEDDS is not clear and concise as it is more on descriptive. So please make a table with details of its current application specifically and another table with details of FDA approved or under clinical trials SEDDS formulation.

In Table 1. Solid SEDDS manufacturing method, can state more on related to technique like polymers used, parameters like temperature, vacuum speed, time and degree of stability etc. 

So please do your best to improve those tables.

Comments on the Quality of English Language

Not Applicable

Author Response

  1. I appreciate so much your comments.
  2. Indicated the revised parts with red color in this revised manuscript.

Reviewer #1

Dear Authors,

Thanks for your contribution. You have compiled a significant information on solidification techniques and applications, however i would like to suggest you for further improvement.

Point 1.  Applications of Solid SEDDS is not clear and concise as it is more on descriptive. So please make a table with details of its current application specifically and another table with details of FDA approved or under clinical trials SEDDS formulation.

In Table 1. Solid SEDDS manufacturing method, can state more on related to technique like polymers used, parameters like temperature, vacuum speed, time and degree of stability etc. 

So please do your best to improve those tables.

Response 1: Thank you for the helpful comments. We revised the manuscript as follows:

Revised “Table 1”. Added two columns—Materials and Critical process parameters—for each technique (polymers/solid carriers and key controllable variables such as temperature, pressure/vacuum, agitation, time).

Added “Table 2” and “Table 3”

Added 12 references

Reviewer 2 Report

Comments and Suggestions for Authors

This review explores Self-Emulsifying Drug Delivery Systems (SEDDS) in detail, particularly their transition from liquid to solid forms. It highlights how liquid SEDDS improve the solubility and bioavailability of many drugs, while discussing their limitations in terms of stability and manufacturing. The article then focuses on solid SEDDS, presenting a comprehensive analysis of solid carrier materials (such as silica, polymers, and carbon-based materials) and solidification techniques (such as spray drying, hot extrusion, and 3D printing). Finally, it discusses advanced applications of solid SEDDS, including controlled release and targeted drug delivery, while highlighting the commercial challenges and future prospects of this innovative technology.

A few minor corrections to be made:
- Replace the numbering 4.3.4 with 4.4
- In Figure 1:
1- Change the position of HP-beta-CD from Polymer-based carriers to Carbohydrate-based materials.
2- Change the cyclodextrin structure, as it is incorrect.

Author Response

  1. I appreciate so much your comments.
  2. Indicated the revised parts with red color in this revised manuscript.

Reviewer #2

This review explores Self-Emulsifying Drug Delivery Systems (SEDDS) in detail, particularly their transition from liquid to solid forms. It highlights how liquid SEDDS improve the solubility and bioavailability of many drugs, while discussing their limitations in terms of stability and manufacturing. The article then focuses on solid SEDDS, presenting a comprehensive analysis of solid carrier materials (such as silica, polymers, and carbon-based materials) and solidification techniques (such as spray drying, hot extrusion, and 3D printing). Finally, it discusses advanced applications of solid SEDDS, including controlled release and targeted drug delivery, while highlighting the commercial challenges and future prospects of this innovative technology.

Point 1.  A few minor corrections to be made:
- Replace the numbering 4.3.4 with 4.4

Response 1: Thanks for the good point. We have revised it as suggested.

Point 2.  In Figure 1:
1- Change the position of HP-beta-CD from Polymer-based carriers to Carbohydrate-based materials.
2- Change the cyclodextrin structure, as it is incorrect.

Response 2: Figure 1 has been revised by relocating HP-β-CD from “Polymer-based carriers” to “Carbohydrate-based materials,” and by correcting the illustration to accurately depict the β-cyclodextrin structure.

Reviewer 3 Report

Comments and Suggestions for Authors

Thank you for the well-structured and comprehensive review on solid SEDDS. The manuscript covers a wide scope, however, there are a few points that need attention to further improve clarity, technical accuracy, and reader usefulness. My specific comments are listed below:

  • Please add a table with marketed products list by including the solid carriers (diluents/polymers) used to solidify the L-SEDDS and any other excipient like stabilizer used to improve the stability.
  • Line 21- all the applications mentioned are for conventional tablets not only for S-SEDDS. Hence, remove the word advanced.
  • Line 38- for successful drug product Please correct it.
  • Line 82- To successfully formulate as SEDDS, Please correct it.
  • Line103- mechanism for maximizing drug oral Please correct it.
  • Section 2.2- Please provide few examples of Surfactants used in SEDDS to improve the reader’s attention.
  • Line 179- Authors mentioned drug loading capacity is one of the limitations. Hence remove it from here.
  • Line 244 to 249- include the adsorption property of the polymeric carriers (solidifying property) along with controlling the drug release, as the manuscript focused on carrier systems for Liquid SEDDS.
  • Write the limitations of each type of solid carriers for screen the best suitable one. This will be helpful for readers to find the best suited carrier for their formulations.
  • Does any carrier will be used in spherical crystallization technology? Or simply precipitating the drug by dissolving in good solvent and precipitating in bad solvent?
  • Line 523 to 554- Based on the requirement, die can be attached to the HME barrel to produce films or filaments or pellets (filament cut into pellet size). To extrude granules, die will be detached from the barrel and extruded the granules. As lipids were used in the formulation composition, processing temperatures will go down than the melting point of drug or glass transition temperature of used polymeric carrier. Please correct these.
  • Also, for HME Liquid SEDDS and solid carrier will be feed into barrel into different zones to extrude the solid SEDDS or semi-solid (waxy) or solid lipids and drug were feed in the first feeding zone after applying the temperature drug will be dissolved in the molten lipid then solid carrier will be added at later zones to extrude out solid SEDDS in a single step process.
  • In section 5.7 if authors could write about soft gelatin capsules (considered as solids) for liquid SEDDS filling and hard gelatin capsules for drugs loaded into solid lipids (stored in refrigerated condition to avoid melting of the solid lipids), it is justifiable. If liquid SEDDS were filled into hard gelatin capsule special equipment (not economical) is required for filling and capsule band sealing. Also, one of the main problems is with physical stability of hard gelatin capsule. Hence correct the paragraphs.
  • Give example for volatile compounds (e.g., some surfactants).
  • Table 1 is already explained in the manufacturing methods sections (5.1 to 5.9). No need to mention them separately here again. Instead, If authors get any information about specific surface area (mentioned in line 317) for other types of carriers please include that in a table format along with maximum oil adsorption capacity.
  • Line 733- Semi permeable or impermeable membrane coating?
  • Section 6.2, Lines 749-759- case studies mentioned are general GRDDS formulations not specific to S-SEDDS.
  • Similarly for Section 6.3 also. Include specific to S-SEDDS formulations, how S-SEEDS will be effective for these applications, including any specific mechanism is helpful in achieving the targeted drug delivery (for example, due to the lower density, SEDDS will remain in stomach and releases the drug, or any chemical or enzyme or receptor mediated interactions exist between lipids and gastric contents/mucous).
  • Looks Section 6.5 is repetition of 5.9.
  • Line 825 to 826- Didn’t understand what authors are talking about.
  • Line 844- Might be Liquid SEDDS. Please correct.
  • In challenges also include liquid loading capacity for solid carriers.

Author Response

  1. I appreciate so much your comments.
  2. Indicated the revised parts with red color in this revised manuscript.

Reviewer #3

Thank you for the well-structured and comprehensive review on solid SEDDS. The manuscript covers a wide scope, however, there are a few points that need attention to further improve clarity, technical accuracy, and reader usefulness. My specific comments are listed below:

Point 1.  Please add a table with marketed products list by including the solid carriers (diluents/polymers) used to solidify the L-SEDDS and any other excipient like stabilizer used to improve the stability.

Response 1:  Added “Table 3”  and Added 12 references

Added “Solid Self-Emulsifying Drug Delivery System (SEDDS) technology ---  the following commercial considerations must be satisfied.”  in line 874-877.

Point 2.  Line 21- all the applications mentioned are for conventional tablets not only for S-SEDDS. Hence, remove the word advanced.

Response 2:  Revised “Furthermore, this review explores applications of solid SEDDS, ---  as well as the key commercial challenges and future perspectives.”  in line 21-23.

Point 3.  Line 38- for successful drug product Please correct it.

Response 3:  Revised “Improving the bioavailability of such drugs has been a persistent challenge for suc-cessful drug products.”  in line 37-38.

Point 4.  Line 82- To successfully formulate as SEDDS, Please correct it.

Response 4:  Revised “To successfully formulate as SEDDS, --- and to optimize their proportions.”  in line 82-83.

Point 5.  Line103- mechanism for maximizing drug oral Please correct it.

Response 5:  Revised “This is a crucial mechanism for maximizing drug oral bioavailability [26].”  in line 102-103.

Point 6.  Section 2.2- Please provide few examples of Surfactants used in SEDDS to improve the reader’s attention.

Response 6:  Added “Representative non-ionic surfactants used in SEDDS --- rapid o/w self-emulsification and stable nanoemulsions.”  in line 126-130.

Point 7.  Line 179- Authors mentioned drug loading capacity is one of the limitations. Hence remove it from here.

Response 7:  Revised “For example, specific solid carriers can offer additional benefits, such as controlling the drug release rate and modulating bioavailability [25, 45].”  in line 181-183.

Point 8.  Line 244 to 249- include the adsorption property of the polymeric carriers (solidifying property) along with controlling the drug release, as the manuscript focused on carrier systems for Liquid SEDDS.

Response 8:  Revised “Polymer-based materials such as hydroxypropyl methylcellulose (HPMC), --- to adsorb/immobilize the liquid SEDDS and to control drug release [64].”  in line 254-257.

Point 9.  Write the limitations of each type of solid carriers for screen the best suitable one. This will be helpful for readers to find the best suited carrier for their formulations.

Response 9:  Thanks for the good point. Added “In particular, silica and silicate-based carriers, as well as inorganic mesoporous materials, ---  be selected based on the characteristics of the target product.” in line 216-222.

Point 10.  Does any carrier will be used in spherical crystallization technology? Or simply precipitating the drug by dissolving in good solvent and precipitating in bad solvent?

Response 10:  Added “Notably, this technique does not require a dedicated solid carrier; --- and facilitate downstream handling.”  in line 414-419.

Point 11.  Line 523 to 554- Based on the requirement, die can be attached to the HME barrel to produce films or filaments or pellets (filament cut into pellet size). To extrude granules, die will be detached from the barrel and extruded the granules. As lipids were used in the formulation composition, processing temperatures will go down than the melting point of drug or glass transition temperature of used polymeric carrier. Please correct these.

Response 11:  Revised “For solid SEDDS, the feed strategy can be configured in a single step --- thereby reducing the risk of thermal degradation.”  in line 542-553.

Point 12.  Also, for HME Liquid SEDDS and solid carrier will be feed into barrel into different zones to extrude the solid SEDDS or semi-solid (waxy) or solid lipids and drug were feed in the first feeding zone after applying the temperature drug will be dissolved in the molten lipid then solid carrier will be added at later zones to extrude out solid SEDDS in a single step process.

Response 12:  Revised “For solid SEDDS, the feed strategy can be configured in a single step --- thereby reducing the risk of thermal degradation.”  in line 542-553.

Point 13.  In section 5.7 if authors could write about soft gelatin capsules (considered as solids) for liquid SEDDS filling and hard gelatin capsules for drugs loaded into solid lipids (stored in refrigerated condition to avoid melting of the solid lipids), it is justifiable. If liquid SEDDS were filled into hard gelatin capsule special equipment (not economical) is required for filling and capsule band sealing. Also, one of the main problems is with physical stability of hard gelatin capsule. Hence correct the paragraphs.

Response 13:  Thanks for the good point. Rivised “While conceptually straightforward, direct liquid filling into hard gelatin capsules ---  Nevertheless, there are potential drawbacks.” in line 623-628.

Point 14.  Give example for volatile compounds (e.g., some surfactants).

Response 14:  Rivised “Additionally, volatile compounds (e.g., ethanol, isopropanol, acetone, ethyl acetate) --- which may affect the final composition of the formulation [148].” in line 665-667.

Point 15.  Table 1 is already explained in the manufacturing methods sections (5.1 to 5.9). No need to mention them separately here again. Instead, If authors get any information about specific surface area (mentioned in line 317) for other types of carriers please include that in a table format along with maximum oil adsorption capacity.

Response 15:  Regarding Table 1, we note the reviewer's concern, but we were unable to delete it as another reviewer requested that the included information be retained and revised.

Furthermore, we kindly clarify that the maximum oil adsorption capacity you inquired about is often not reported in other studies, making it challenging to standardize and compile in a comprehensive table format.

Point 16.  Line 733- Semi permeable or impermeable membrane coating?

Response 16:  Rivised “For example, a solid self-emulsifying osmotic pump tablet  --- by a semipermeable membrane coating and laser-drilling an orifice.” in line 759-762.

Point 17.  Section 6.2, Lines 749-759- case studies mentioned are general GRDDS formulations not specific to S-SEDDS.

Response 17:  Rivised “Floating and gastroretentive S-SEDDS formulations --- enhance bioavailability through prolonged gastric residence and controlled release.” in line 778-791.

Added 3 references.

Point 18.  Similarly for Section 6.3 also. Include specific to S-SEDDS formulations, how S-SEEDS will be effective for these applications, including any specific mechanism is helpful in achieving the targeted drug delivery (for example, due to the lower density, SEDDS will remain in stomach and releases the drug, or any chemical or enzyme or receptor mediated interactions exist between lipids and gastric contents/mucous).

Response 18:  Rivised “Mucoadhesive polymers, such as chitosan and thiolated polyacrylic acid deriva-tives, --- by contributing to sustained drug release, enhanced bioavailability, and targeted drug delivery.” in line 793-808.

Added 3 references.

Point 19.  Looks Section 6.5 is repetition of 5.9.

Response 19:  Rivised “As previously discussed in Section 5.9, Solid SEDDS enables personalized medicine through integration with 3D printing technology.” in line 842-843.

Point 20.  Line 825 to 826- Didn’t understand what authors are talking about.

Response 20: We apologize for the lack of clarity in our initial wording. We have revised the relevant text accordingly to ensure greater clarity in the manuscript.

Rivised “A promising approach to overcome this problem is to adsorb SEDDS onto carriers with high surface area and high surface porosity, thereby increasing drug loading. This ap-proach offers the potential to maximize the final drug content in solid SEDDS.” in line 862-864.

Point 21.  Line 844- Might be Liquid SEDDS. Please correct.

Response 21: Rivised “Liquid SEDDS have led to innovation by solving one of the biggest challenges in the pharmaceutical industry: the solubility and bioavailability of poorly water-soluble drugs.” in line 884-885.

Point 22.  In challenges also include liquid loading capacity for solid carriers.

Response 23:  Thanks for the good point. Rivised “Challenges such as the toxicity of high-concentration surfactants, formulation limitations for drugs with low lipid solubility, liquid loading capacity for solid carriers, and the technical complexity of manufacturing scale-up require continuous research and innovation.”  in line 899-902.

Round 2

Reviewer 1 Report

Comments and Suggestions for Authors

Dear Authors,

Thank you so much for your revision. 

Author Response

We thank the reviewer for their valuable comments and time in reviewing our manuscript. 

Reviewer 3 Report

Comments and Suggestions for Authors

The authors addressed most of the given comments except for two. To strengthen the manuscript, please include,

  • Add a table with the marketed products list by including the solid carriers (diluents/polymers) used to solidify the L-SEDDS and any other excipient, like a stabilizer or antioxidant or something else used to improve the stability of the drug product (S-SEDDS).
  • The limitations of each type of solid carrier for screening the best suitable one. This will be helpful for readers to find the best-suited carrier for their formulations.

Author Response

  1. I appreciate so much your comments.
  2. Indicated the revised parts with red color in this revised manuscript.

Reviewer #3

The authors addressed most of the given comments except for two. To strengthen the manuscript, please include,

Point 1.  Add a table with the marketed products list by including the solid carriers (diluents/polymers) used to solidify the L-SEDDS and any other excipient, like a stabilizer or antioxidant or something else used to improve the stability of the drug product (S-SEDDS).

Response 1:  Thank you for your valuable suggestion.

We have revised the manuscript and added a new Table 3, which includes a list of currently marketed SEDDS-related products and those undergoing clinical trials. We have diligently attempted to address the specific details you requested by conducting an extensive search of the literature and patents. However, we found it difficult to locate the information regarding the specific solid carriers (diluents/polymers) and stability-improving excipients used in these commercial products. We believe that the precise details of the solid carriers and stabilizers often constitute proprietary knowledge (core know-how) of the pharmaceutical companies, and consequently, this information is not widely disclosed for commercial products. Therefore, we kindly ask for your understanding that we are limited to mentioning the need for further investigation into these specific points within the discussion section of the manuscript.

Added “4) Solidification and stabilization: To improve the production and stability of solid SEDDS, a comprehensive review of the use of solid carriers (diluents/polymers), stabi-lizers, and antioxidants is required for successful commercialization.”  in line 897-900.

Point 2.  The limitations of each type of solid carrier for screening the best suitable one. This will be helpful for readers to find the best-suited carrier for their formulations.

Response 2:  Thank you for your valuable review. We have addressed the limitations of each solidification carrier you mentioned in the revised manuscript.

Added “Silica and silicate-based materials are --- they are incorporated in large quantities due to potential drug binding.”, “Polymer-based materials, as mentioned above, --- which delays re-emulsification and leads to system instability.”, “In general, inorganic mesoporous-based materials have --- crucial for maximizing the performance of solid SEDDS.”, and “As mentioned earlier, carbohydrate-based materials ---- carbohydrate-based materials can offer versatility and biocompatibility in the design of solid SEDDS.”  in line 250-252, line 289-293, line 374-378, and line 405-408.
